# Wide Temperature Range and Low Temperature Drift Eddy Current Displacement Sensor Using Digital Correlation Demodulation

**DOI:** 10.3390/s23104895

**Published:** 2023-05-19

**Authors:** Tianxiang Ma, Yuting Han, Yongsen Xu, Pengzhang Dai, Honghai Shen, Yunqing Liu

**Affiliations:** 1Information and Communication Engineering, School of Electronic and Information Engineering, Changchun University of Science and Technology, Changchun 130022, China; 2Key Laboratory of Airborne Optical Imaging and Measurement, Changchun Institute of Optics, Fine Mechanics and Physics, Chinese Academy of Sciences, Changchun 130033, China; 3University of Chinese Academy of Sciences, Beijing 100039, China

**Keywords:** eddy current sensor, temperature drift, digital demodulation, double correlation demodulation

## Abstract

Conventional eddy-current sensors have the advantages of being contactless and having high bandwidth and high sensitivity. They are widely used in micro-displacement measurement, micro-angle measurement, and rotational speed measurement. However, they are based on the principle of impedance measurement, so the influence of temperature drift on sensor accuracy is difficult to overcome. A differential digital demodulation eddy current sensor system was designed to reduce the influence of temperature drift on the output accuracy of the eddy current sensor. The differential sensor probe was used to eliminate common-mode interference caused by temperature, and the differential analog carrier signal was digitized by a high-speed ADC. In the FPGA, the amplitude information is resolved using the double correlation demodulation method. The main sources of system errors were determined, and a test device was designed using a laser autocollimator. Tests were conducted to measure various aspects of sensor performance. Testing showed the following metrics for the differential digital demodulation eddy current sensor: nonlinearity 0.68% in the range of ±2.5 mm, resolution 760 nm, maximum bandwidth 25 kHz, and significant suppression in the temperature drift compared to analog demodulation methods. The tests show that the sensor has high precision, low temperature drift and great flexibility, and it can instead of conventional sensors in applications with large temperature variability.

## 1. Introduction

An eddy current sensor is a contactless sensor [1]. When a high-frequency AC signal flows through the coil of the sensor, an alternating magnetic field is generated around the coil. An eddy current is induced within a nearby conductor by the alternating magnetic field, which generates another magnetic field in the opposite direction of the coil. The interaction between the magnetic field of the coil and the eddy current changes the AC impedance of the sensor coil. The distance between the detector coil and the metal being tested is proportional to the coupling strength of the magnetic field, so the displacement of the target conductor can be obtained by measuring the AC impedance [2,3]. However, the disadvantage of an eddy current sensor is that there is a large temperature drift. Because the eddy current sensor is essentially an impedance measurement circuit, the coil of the sensor, and the impedance and resistivity of the metal to be measured, will change with temperature, which will affect the measurement [4,5,6].

Scholars in the field of overcoming the temperature drift of eddy current sensors typically employ two methods. The first approach is data processing, which leverages prior knowledge and algorithms to post-calibrate the sensor’s output data and enhance its precision. For example, a temperature compensation method based on binary regression was proposed by Lei et al. to reduce the temperature drift effect, which minimized the maximum relative error from 169.08% to 9.13% [7].

He et al. proposed an external compensation approach by applying mathematical fitting to realize temperature drift compensation, and it improved the temperature compensation accuracy to 0.25% [8].

The second method involves incorporating a self-compensation module into the design of the eddy current sensor, thereby using circuit design to achieve temperature compensation and enhance the accuracy of the output. For example, Li and Ding designed a new displacement eddy current sensor for high-voltage applications that incorporated a simple temperature drift compensation method through the use of a non-inductive compensation coil, which reduced temperature drift from 12% to 0.7% within the temperature range of 20 to 90 °C [9].

The temperature drift of the sensor impedance due to the temperature change in the probe coil and the measured target environment has been thoroughly discussed. Wang et al. concluded that the coil’s resistance was the primary cause of the temperature drift. They proposed a new integrated self-temperature-compensation method that leveraged signal processing of the sensing coil to achieve an ultra-low temperature drift of 4 nm/°C [10].

Wang and Feng also designed an innovative bridge circuit and combined self-calibration and impedance-based signal processing methods to significantly reduce the temperature drift of the eddy current sensor by two orders of magnitude. The thermal drift of the sensor was only 2.6 nm/°C [5].

Zheng et al. designed a special mechanical structure and a corresponding sensor compensation circuit to address the exponential temperature drift caused by an air gap, metal material, and probe coils in high-temperature applications. The proposed compensation method reduced the exponential hysteresis temperature drift errors by 79.65% for a temperature rise of 80 K and by 85.04% for a temperature rise of 320 K [11].

Based on a constant-current circuit and a temperature coefficient parameter method, Wang et al. designed a displacement sensor with a temperature compensation function, which achieved a temperature drift rate of ±0.96 µm/°C within the temperature range of 40 °C to 55 °C [12].

The data post-calibration method can enhance the accuracy of the data, but the procedure is complex and necessitates individual data collection and processing for each sensor. The use of compensation circuits in sensors can often improve accuracy and reduce temperature drift, but it may also limit the measurement range. Additionally, the impact of the temperature characteristics of each electronic component in the demodulation circuit has not been thoroughly evaluated and compensated. Herein, a method suitable for reducing the temperature drift of the sensor in a wide-range temperature environment is proposed, significantly improving the temperature stability of the eddy current displacement sensor. The proposed method utilizes the sensor structure of differential dual probes to offset the traditional temperature drift caused by the coil, the target metal, and the cable. Moreover, high-speed analog-to-digital sampling is performed on the demodulated signal, and the double-correlation demodulation algorithm is employed in the digital processor to process the amplitude and phase of the sensor. The sensor designed based on the above ideas achieved a maximum static stability accuracy of ±1 Digital Number (DN) with a system bandwidth of up to 25 kHz. The temperature drift rate was found to be ±0.00137% FS/°C when the temperature was increased from −40 °C to +50 °C, making this method flexible, reliable, and convenient for applications with a wide temperature range.

## 2. Measurement

### 2.1. Differential Eddy Current Sensor

The basic structure of the eddy current sensor is a parallel resonant LC circuit. The principal circuit blocks of the sensor are shown in Figure 1. The driving signal $U_{i}$, with constant frequency and constant amplitude, is input to the parallel resonant circuit, consisting of a sensing coil $L_{1}$ and a capacitor $C_{1}$. When the distance between the sensing coil and the metal being tested changes, the coefficient M of mutual inductance between $L_{1}$ and $L_{2}$ will change, which will affect the output signal $U_{0}$. The distance information between the sensor coil and the measured object has an approximately linear relationship with the amplitude of the output signal.

The relationship between the excitation signal and the output signal of the sensor is shown in Equation (1).
(1)$$U_{0}=U_{i}\frac{R_{2}+R_{L}+j\omega \left(L_{1}-L_{L}\right)}{R_{2}+R_{L}+j\omega \left(L_{1}-L_{L}\right)+R_{1}\left[1-\omega^{2}C_{1}\left(L_{1}-L_{L}\right)\right]+j\omega C_{1}R_{1}(R_{2}+R_{L})}$$
where
(2)$$R_{L}=\frac{{\left(M\omega \right)}^{2}R_{3}}{{\left(R_{3}\right)}^{2}+{\left(\omega L_{2}\right)}^{2}},$$
(3)$$L_{L}=\frac{{\left(M\omega \right)}^{2}L_{2}}{{\left(R_{3}\right)}^{2}+{\left(\omega L_{2}\right)}^{2}}.$$

The amplitude and the frequency ω of the excitation signal $U_{i}$, the coil’s equivalent loss resistance $R_{2}$, the coil’s inductance $L_{1}$, the parallel resonant capacitor $C_{1}$, the voltage divider resistor $R_{1}$, the resistance $R_{3}$, and the inductance $L_{2}$ introduced by the target conductor will all affect the output of the sensor. *M* is the coefficient of mutual induction. The equivalent resistance and inductance of the coil are determined by the mutual inductance coefficient *M*. This coefficient varies with the distance between the coil and the metal conductor. By measuring the change in the output voltage resulting from the variation in the sensor coil impedance, a mathematical relationship between the distance and the mutual inductance coefficient can be established. The circuit shows that the influence of $L_{2}$ and $R_{3}$ on the output in tests is equivalent to the influence of $L_{1}$ and $R_{2}$, so Equation (1) can be rewritten as
(4)$$U_{0}=U_{i}\frac{R_{2}+j\omega L_{1}}{R_{2}+j\omega L_{1}+R_{1}\left[1-\omega^{2}C_{1}L_{1}\right]+j\omega C_{1}R_{1}R_{2}}.$$

As demonstrated in Equation (4), the output of the circuit is contingent upon the change in impedance when the input signal remains constant. By setting the circuit parameters in the equation equal to *K*, Equation (5) can be derived.
(5)$$K=\frac{R_{2}+j\omega L_{1}}{R_{2}+j\omega L_{1}+R_{1}\left[1-\omega^{2}C_{1}L_{1}\right]+j\omega C_{1}R_{1}R_{2}}$$

Two sensor coils with identical structures were selected to form a differential sensing system that eliminated the common-mode interference caused by the temperature drift of the coils, the temperature drift due to the metal resistivity of the target, and the temperature drift due to cable impedance. The arrangement is shown in Figure 2.

Assuming that the corresponding circuit parameters of the two sensor probes are equal, a linear function relating the circuit parameters and the output differential voltage of the two sensor probes can be derived, as shown in Equation (6).
(6)$$U_{0\_diff}=U_{R}-U_{L}=\left(K_{R}-K_{L}\right)U_{i}$$
where $U_{i}$ is a constant input signal and $K_{R}$ and $K_{L}$ represent the equivalent impedance parameters of the two probes at the current position in the system, respectively.

### 2.2. Structure of the Digital Processing Circuit

The signal output by the temperature-compensated crystal oscillator is a 500 kHz driving signal. After the signal passes through the resonant network, it drives the two differential probes. The signal output from the probe first enters the emitter follower circuit for amplification. The two amplified signals are subtracted at the next stage to obtain the differential-mode signal measured by the probe, which is input to the high-speed ADC after the differential buffer to complete the digital acquisition. The signal that triggers the FPGA and enters the coils is from same source, which is generated by the buffer circuit. The digital signal processing circuit block diagram is shown in Figure 3.

### 2.3. Digital Demodulation

The input signal is:(7)$$x\left(t\right)=s_{in}\left(t\right)+n\left(t\right),$$
where $s_{in}\left(t\right)$ is the signal being tested, and $n\left(t\right)$ is the noise signal in the system. The reference signal is
(8)$$y\left(t\right)=s_{ref}\left(t\right)$$

The reference signals for demodulation, which are the discrete sine and discrete cosine signals, are generated by the COE core embedded in the FPGA. If the system signal to be measured is correlated with the reference signal and the reference signal is not correlated with the noise signal, then the input signal is delayed, multiplied, integrated and averaged to obtain the cross-correlation function of the system [13,14], as shown in Equation (9):(9)$$\begin{array}{l} R_{xy}\left(\tau \right)=\underset{t\to \infty }{\lim}\frac{1}{2T}\int_{-T}^{T}x\left(t\right)y\left(t-\tau \right)dt\\ \;\;\;\;\;\;\;\;\;\;\;\;=\underset{T\to \infty }{\lim}\left[\frac{1}{2T}\int_{-T}^{T}s_{in}\left(t\right)s_{ref}\left(t-\tau \right)dt+\frac{1}{2T}\int_{-T}^{T}n\left(t\right)s_{ref}\left(t-\tau \right)dt\right]\\ \;\;\;\;\;\;\;\;\;\;\;\;\;=R_{s_{in}s_{ref}}\left(\tau \right)+R_{ns_{ref}}\left(\tau \right) \end{array}$$

Given the assumption that the reference signal is correlated with the measured signal, and assuming that the noise signal in the system is Gaussian white noise with a low-frequency band limit of normal distribution and a mean value of 0 [15], it can be deduced that
(10)$$R_{xy}\left(\tau \right)=R_{s_{in}s_{ref}}\left(\tau \right).$$

The output $U_{0}$ of the associated demodulator is
(11)$$U_{0}=\frac{U_{s}U_{r}}{2}\cos\varphi .$$
$U_{s}$ and $U_{r}$ in Equation (11) are, respectively, the amplitudes of the measured signal and the reference signal, and φ is the phase difference between the two signals.

If the reference signal is transformed into a sinusoidal signal with the same frequency as the measured signal, the same reasoning will produce
(12)$$U_{0}=\frac{U_{s}U_{r}}{2}\sin\varphi .$$

Combining Equations (11) and (12) gives the output expression of the analog double correlation demodulator [16]:(13)$$U_{s}=\frac{2\times \sqrt{{\left(\frac{U_{s}U_{ref}}{2}\cos\varphi \right)}^{2}+{\left(\frac{U_{s}U_{ref}}{2}\sin\varphi \right)}^{2}}}{U_{ref}}.$$

In the digital system, the sampling frequency of the analog signal is $f_{s}$, and N data points are collected in each period. The discrete expression of the analog signal can be written as
(14)$$x_{s}\left(k\right)=U_{s}\cos\left(\frac{2\pi k}{N}\right).$$

The corresponding reference sine and cosine signal expressions are
(15)$$y_{\cos}\left(k\right)=U_{\cos}\cos\left(\frac{2\pi k}{N}+\theta \right),$$
(16)$$y_{\sin}\left(k\right)=U_{\sin}\sin\left(\frac{2\pi k}{N}+\theta \right).$$

The cross-correlation functions of the signal and the reference sine signal and reference cosine signal are
(17)$$R_{s\cos}=\frac{1}{M}\sum_{k=0}^{M-1}x_{s}\left(k\right)y_{\cos}\left(k\right),$$
(18)$$R_{s\sin}=\frac{1}{M}\sum_{k=0}^{M-1}x_{s}\left(k\right)y_{\sin}\left(k\right).$$

*M* in the equation is the accumulated number of cycles, it can be seen that
(19)$$R_{s\cos}=\frac{U_{s}U_{\cos}}{2}\cos\theta ,$$
(20)$$R_{s\sin}=\frac{U_{s}U_{\sin}}{2}\sin\theta .$$

Combining Equations (19) and (20), we obtain [17,18]
(21)$$U_{s}=\frac{2\times \sqrt{R_{s\sin}^{2}+R_{s\cos}^{2}}}{U_{ref}}.$$

## 3. System Error Analysis

### 3.1. Influence of Accumulation Cycle and Sampling Frequency on Output Accuracy

The essential task of the cross-correlation function in the digital correlation demodulation algorithm is to calculate the cumulative averages of the input signal and the reference signal, which makes it equivalent in function to a low-pass filter. Because the noise item has no correlation with the reference signal, the cross-correlation function is 0. Its output is related only to the input signal and the local reference signal. However, the elimination of the noise term in the digital correlation demodulation algorithm presupposes that the averaging time is of adequate length, as data accuracy can be compromised when the algorithm does not accumulate enough data.

Figure 4 illustrates the effect of extending the accumulation period. The initial amplitudes of both the input signal and the reference signal of the same frequency are 1, and the number of accumulation cycles in the algorithm is gradually increased. As up to 15 cycles are accumulated, the output accuracy of the system greatly increases. As 15–50 cycles are accumulated, accuracy increases by up to 0.02%. As 50–100 cycles are accumulated, accuracy increases by <0.01%. As 100–500 cycles are accumulated, accuracy increases by <0.0005%. In a digital system, the accumulation period is inversely proportional to the system output frequency. Choosing a suitable accumulation period can therefore balance the data-updating frequency with the degree of accuracy.

It can be seen from Equation (14) that the sampling frequency of the ADC is also a factor that influences output accuracy. A high sampling frequency obtains more discrete points in one cycle, and more discrete points can accelerate the convergence of the algorithm as follows. We set the number of sampling points to the four values 160, 320, 480 and 640, and plotted the calculated values corresponding to each of the four values for 1–50 accumulation cycles, as shown in Figure 5. From the figure, we can see the influence of the sampling cycle on the convergence speed and the accuracy of the algorithm.

It can be seen from Figure 5 that when the accumulated length is 15 cycles and the number of sampling points increases from 160 to 640, the accuracy of the output amplitude increases by 0.04%. However, as the number of accumulated cycles increases, the influence of the number of sampling points on the accuracy gradually decreases.

The sampling frequency of the ADC in the system cannot be increased infinitely as it is limited by the performance of the fabricated ADC. As an example, if the sampling frequency of an ADC with a maximum sampling frequency of 250 MHz is set to 240 MHz, then an input signal of 500 kHz will provide 480 sampling points in one cycle.

### 3.2. Influence of Frequency Stability on Accuracy

Another factor that affects system accuracy is the stability of the input signal frequency. The frequencies of the data signal and the reference signal are required to be equal and strictly constant. If the two frequencies are not equal, the accuracy of the output signal will be greatly affected.

Correlation-based demodulation requires that the frequency difference between the reference signal and the measured signal should be 0. The reference frequency is fixed, while the measured frequency may vary within a small range around the reference frequency. Let the reference signal be
(22)$$x_{r}\left(t\right)=U_{r}\cos\omega_{0}t,$$
and let the measured signal be
(23)$$x_{s}\left(t\right)=U_{s}\cos\omega_{s}t.$$

Assuming that there is no frequency difference between the two signals, the result of multiplying the two signals is
(24)$$\begin{array}{l} x_{r}\left(t\right)x_{s}\left(t\right)=U_{s}U_{r}\cos\omega_{0}t\cos\omega_{s}t\\ \;\;\;\;\;\;\;\;\;\;\;\;\;\;\;\;\;\;=\frac{U_{s}U_{r}}{2}\cos\left[\left(\omega_{s}-\omega_{0}\right)t\right]+\frac{U_{s}U_{r}}{2}\cos\left[\left(\omega_{s}+\omega_{0}\right)t\right] \end{array}$$

After multiplication, the two frequency components are obtained. The structure of the digital demodulation system is such that after multiplying the frequency, the components will be attenuated by the low pass filter, and thus the output of the demodulator is the difference in frequency between the signal components. The amplitude–frequency characteristic formula of the low pass filter is
(25)$$A\left(f\right)=\left|H\left(f\right)\right|=\frac{1}{{\sqrt{1+\left(2\pi f\tau \right)}}^{2}}.$$

When f=0, i.e., when $\omega_{s}-\omega_{0}=0$, $A\left(f\right)$ reaches its maximum value. When there is a difference between the measured signal and the reference signal, the correlation-based demodulation output amplitude will decrease.

In Equation (21), parameters such as the number of sampling points, the amplitude of the reference signal (which is set equal to 1), and the accumulation period remain constant. Only the frequency of the sampled signal is varied, with each sampling point being changed by 5 parts per million (ppm). By doing so, the relationship between frequency variation and output accuracy can be determined.

Figure 6 shows the influence of temperature drift on the output accuracy of the measured signal. When the temperature drift of the measured signal reaches 10 ppm, the corresponding output error will be in the order of 0.01%, and when the temperature drift reaches 50 ppm, the system output error will be in the order of 0.1%.

### 3.3. Influence of Other Factors

It can be seen from Equation (1) that the amplitude of the driving signal is proportional to the amplitude of the output signal when all other factors in the system are held constant. The amplitude of the driving signal is determined by the amplitude of the positive voltage power supply. The output amplitude of the power supply will change with temperature, and when the temperature changes, the amplitude of the driving signal will also change. The linear relationship between the driving signal and the output signal ensures that the output signal will eventually be affected by temperature changes. Figure 7 shows that the output curve of a typical LDO power supply drifts as the temperature changes [19]. When the temperature changes from −40 °C to +50 °C, the amplitude of the power supply changes by 0.03 V. The 16-bit high-speed ADC, with a full-scale range of 2.5 V, will cause a change of 787 DN, although in the differential system we developed, the driving signals of the two sensor probes have the same trend and magnitude of amplitude, and the common mode interference is eliminated after subtraction. However, due to the nonlinearity of sensor factors, the differential signal obtained by the system will also be affected by the change in amplitude.

In addition, due to the differential circuit structure of the system, there will be temperature-drift effects on the sensor coil impedance, the resistivity of the conductor being tested, the distributed parameter impedance of the sensor coaxial cable, and the cache op-amp, which can all be recognized as common-mode interference and so rejected by the differential system.

## 4. Experimental Verification

In this section, we describe the construction of an experimental device to test the performance of the sensor and the experimental testing of the sensor. Testing included static stability testing, linearity testing, resolution and sensitivity testing, and temperature-stability testing.

### 4.1. Experimental Device Construction

Figure 8 shows that the experimental device consisted principally of a one-dimensional displacement-adjustment platform, a sensor probe fixing structure, a driving mechanism, and a laser autocollimator.

The process for setting up the test equipment was as follows. One sensor probe was fixed to the mount. Using the position of the sensor as the starting point, the laser autocollimator was used to adjust the position of the metal plate being tested with respect to the sensor, and when the desired distance had been set, the other sensor probe was affixed. The metal plate being tested was then centralized between the two probes. In the measurement experiment, we set the sensor–plate distances to 2.5 mm. The model of the laser interferometer probe used in this experiment was OMRON ZW-S20, which had a measurement accuracy of 20 nm and a measurement range of 20 ± 1 mm. The metal specimen to be tested was mounted on a micro-displacement platform, specifically the M-UMR8.25 model. The micro-displacement platform was equipped with a differential micrometer, which provided an accuracy of 0.1 μm.

### 4.2. Drive Signal and Resonance Signal Measurement

The clock signal after frequency division was input to the sensor coil, and the differential probe returned carrier signals with the same frequency but with different amplitudes, according to the difference in inductive impedance. One signal was subtracted from the other to produce the signal that was to be demodulated. In Figure 9, the red carrier signal was subtracted from the blue signal to give the green signal that was demodulated, and the three signals had the same frequency.

### 4.3. Static Stability, Linearity Measurement

#### 4.3.1. Static Stability Test

We maintain the stability of the object under examination and vary the accumulation cycle in the algorithm to analyze the output data of the system. Figure 10 shows the static stability of the system. This is consistent with the conclusions of the previous analysis. The accuracy of the system output was proportional to the number of accumulation cycles, and as the number of accumulation cycles increased, the convergence of the calculated points on the *Y*-axis increased. However, the number of accumulation cycles was inversely proportional to the speed of updating the demodulation data. In actual use, compromises have to be made according to the operating conditions.

The variance and standard deviation of the data for the three numbers of accumulation periods are shown in Table 1.

It can be seen from Table 1 that as the number of accumulation periods increased, the range of the distribution of the output amplitude gradually decreased, which is consistent with the conclusion drawn in Section 3.

#### 4.3.2. Repeatability Error, Nonlinearity Linearity Test, and Accuracy Test

The amplitude of the input analog signal was adjusted to approach the full range of the ADC. The differential micrometer was used to move the target position within a range of ±2.5 mm with a single-step displacement of 500 μm. The laser interferometer was utilized to ensure the positioning accuracy of each single-step movement, and the output signal was collected at each point. The repeatability error of the sensor is a measure of the consistency of its output characteristics with multiple movements in the same direction and range. The maximum repeatability error’s expression is [20]
(26)$$e_{z}=\pm \frac{3\sigma }{U_{Fs}}\times 100\%.$$

Figure 11 displays that the repeatability error is ±0.62%.

We used the data with the largest error for nonlinearity calculations, which was calculated as [21]
(27)$$e_{r}=\frac{\Delta D_{\max}}{D_{FS}}.$$

Figure 11 shows that the nonlinearity of the system was 0.68% and that the correlation coefficient between the variables was 0.9999.

The sensor measurement error was calculated using Equation (28).
(28)$$S_{d}=\sqrt{\frac{\sum_{i=1}^{n}\left(s_{i}-s_{std}\right)}{n}}$$
where $s_{i}$ is the measured value, $S_{std}$ is the theoretical value caused by platform displacement, and n is the number of times the measurement was changed. $S_{d}$ indicates the size of the measurement error between the real value and the measured value. A lower value of $S_{d}$ indicates greater measurement accuracy. Using the test data, we obtained $S_{d}\;$ = 99.7.

### 4.4. Resolution and Sensitivity

When the measured target was kept fixed, 500 points were intercepted from the stable output signal of the system to determine the sensor’s resolution. The measurement accuracy of the sensor was calculated using Equation (29); sensitivity was the data resolution error [22]. The range of the high-speed ADC was 2.5 V, and we calculated the sensitivity of the sensor with a range of 2.5 mm as an example.
(29)$$R_{es}=\frac{\sigma }{S}$$

Using a data oscillation amplitude of ±10 DN at static stability, and from Figure 12, we calculated the measurement resolution of the detector using [23]
(30)$$R_{es}=\frac{0.76mV}{1V/mm}=760nm$$

### 4.5. Temperature Stability

The digital demodulation board, differential probe, and probe-fixing device were integrated into the temperature experiment box, as shown in Figure 13.

The differential sensor probe was adjusted and put into the temperature experiment box together with the circuit board. System output was recorded every 10 °C from −40 °C to +50 °C, and the temperature was maintained for one hour at each test point. The experiment was repeated five times, as shown in Figure 14.

We used the data with the largest error for calculations, and the sensor temperature drift was calculated as
(31)$$T_{r}=\pm \frac{\frac{T_{t\_drift}}{T_{t\_fullscale}}}{Temperature_{changed}}\times 100\%.$$

It can be seen from Figure 14 that as the temperature changed from −40 °C to +50 °C, sensor output at the same measurement position changed by 81 DN and the output change rate due to temperature drift was 0.12%, and the temperature drift was ±0.00137% FS/°C. This was a significant improvement in temperature drift performance by the sensor. For every 10 °C change in temperature, the temperature drift rate of the output amplitude was 0.029% at the maximum and 0.0004% at the minimum. At room temperature, the rate of change in temperature drift for every 10 °C was <0.01%.

### 4.6. Summary

After the above test, the sensor’s main performance is listed in Table 2.

## 5. Conclusions

In this study, we designed and tested a low-temperature-drift differential-digital-demodulation eddy-current sensor. The basic mathematical model of the differential eddy-current sensor was presented, a digital eddy current signal amplitude demodulation algorithm was developed, and sources of error in the algorithm were analyzed and eliminated. An eddy current sensor test platform using a laser autocollimator was built to test various sensor parameters. The test results show that nonlinearity was 0.68% and the measurement resolution of the detector was 760 nm. The sensor was tested at various temperature points, and the temperature drift curve of the sensor was plotted. From −40 °C to +50 °C, the system temperature drift rate was ±0.00137% FS/°C, and the temperature drift change rate per 10 °C was <0.01%.

## Figures and Tables

**Figure 1 sensors-23-04895-f001:**
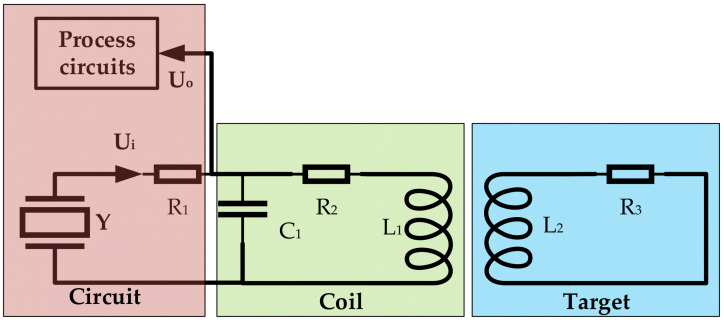
Block diagram of the eddy current sensor.

**Figure 2 sensors-23-04895-f002:**
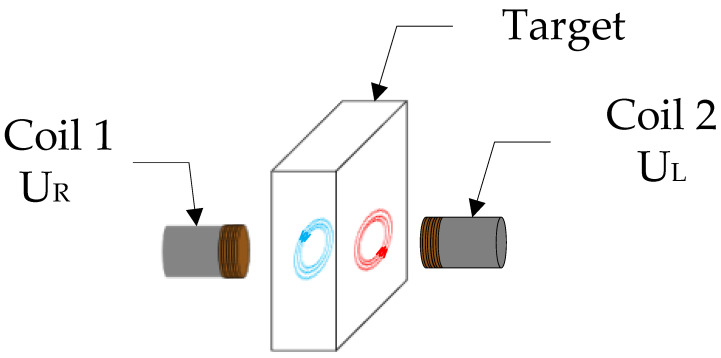
Schematic diagram of coil placement.

**Figure 3 sensors-23-04895-f003:**
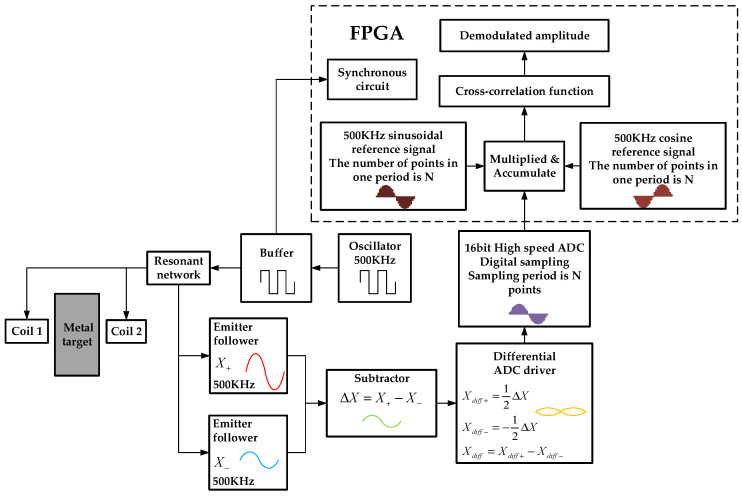
Block diagram of digital signal processing circuit.

**Figure 4 sensors-23-04895-f004:**
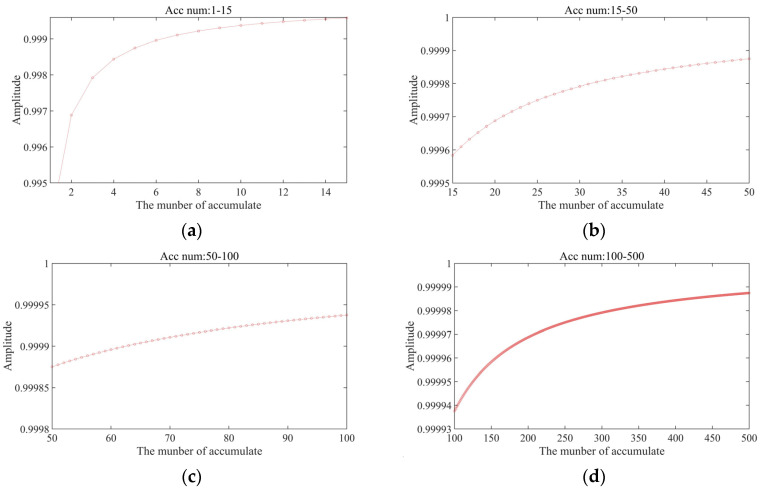
Effect of the accumulation period on output accuracy: (**a**) 1–15 accumulation cycles; (**b**) 15–50 accumulation cycles; (**c**) 50–100 accumulation cycles; (**d**) 100–200 accumulation cycles.

**Figure 5 sensors-23-04895-f005:**
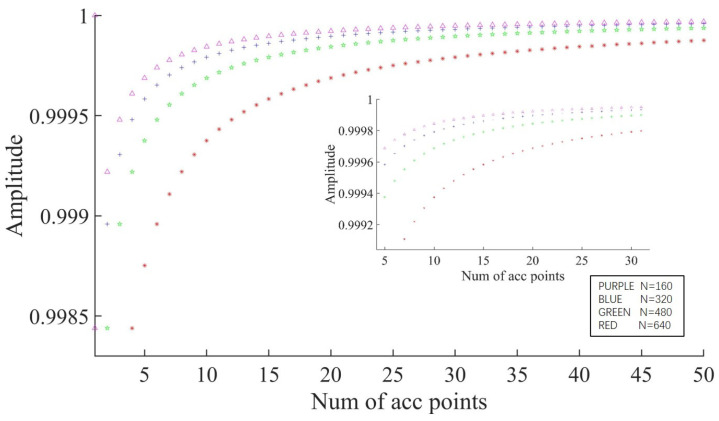
Effects of different numbers of sampling points in 1–50 sampling periods on convergence speed and accuracy (the smaller picture is a partial enlargement of the larger one).

**Figure 6 sensors-23-04895-f006:**
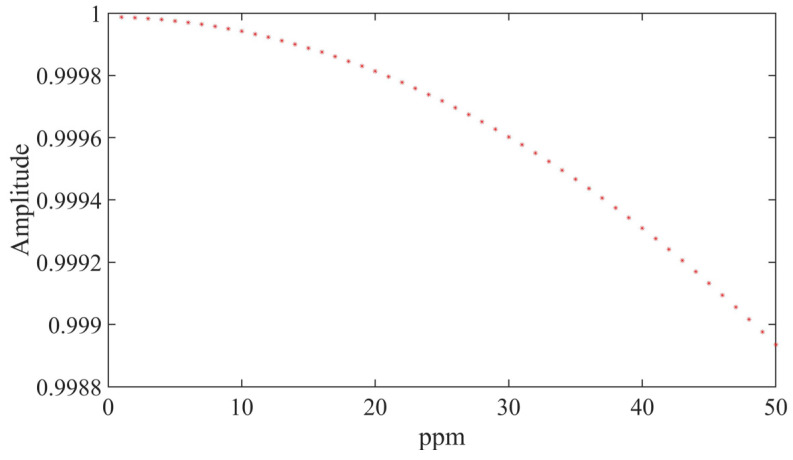
Influence of temperature drift on the output accuracy of the measured signal.

**Figure 7 sensors-23-04895-f007:**
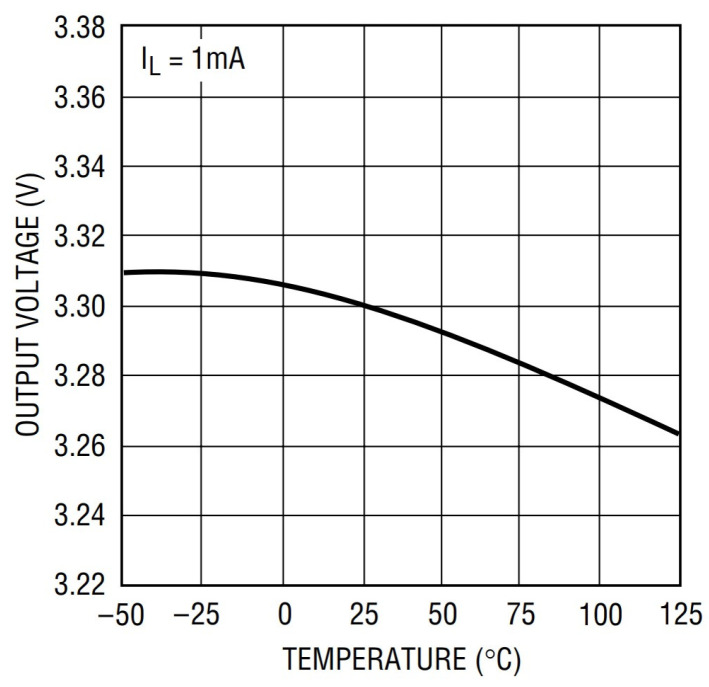
Effect of temperature on power supply output voltage.

**Figure 8 sensors-23-04895-f008:**
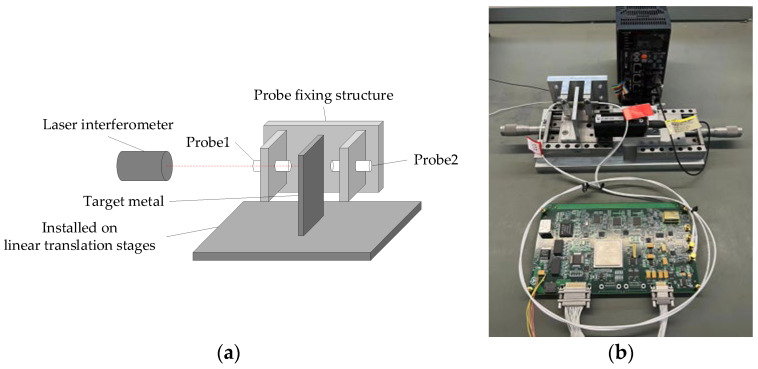
Experimental device: (**a**) schematic diagram of the experimental device; (**b**) physical configuration of the experimental device.

**Figure 9 sensors-23-04895-f009:**
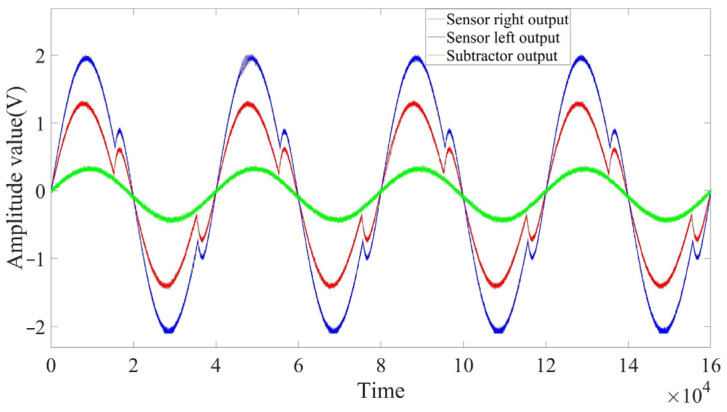
The two carrier signals output by the circuit and the difference signal to be demodulated.

**Figure 10 sensors-23-04895-f010:**
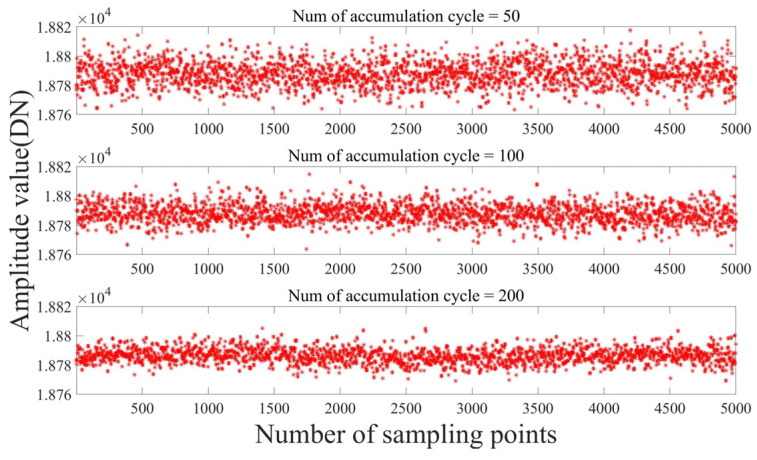
The influence of the number of accumulation cycles on output accuracy.

**Figure 11 sensors-23-04895-f011:**
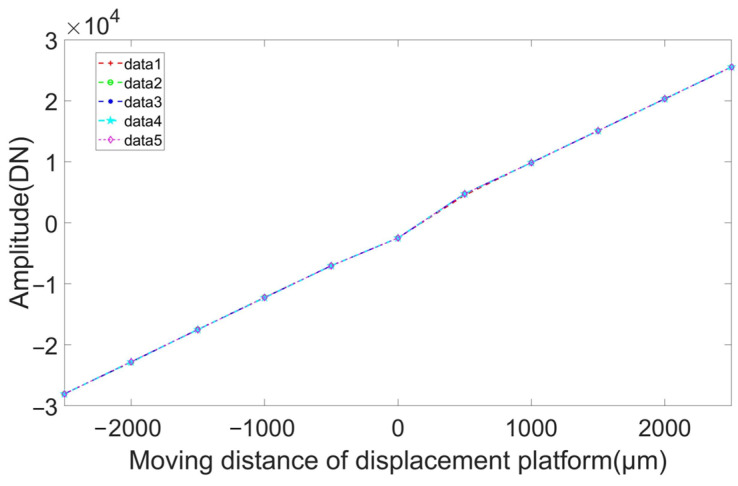
Detector linearity test results.

**Figure 12 sensors-23-04895-f012:**
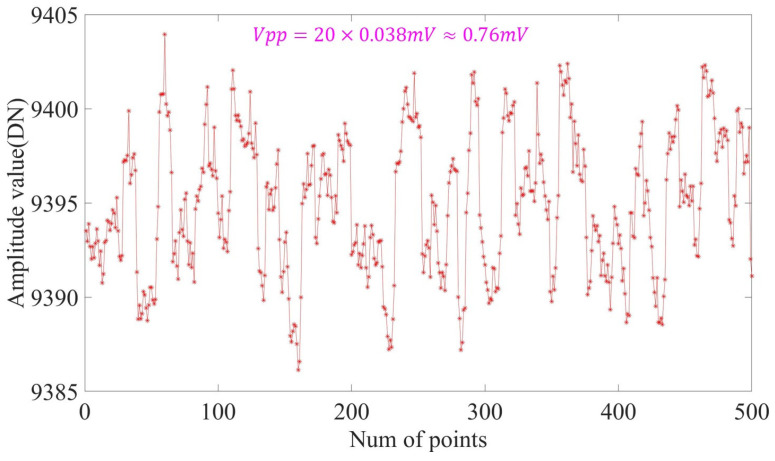
Static stability accuracy of the sensor after demodulation.

**Figure 13 sensors-23-04895-f013:**
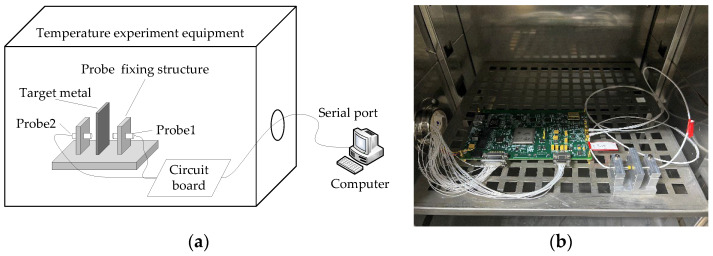
Schematic diagram and physical picture of the experimental device. (**a**) Experimental schematic; (**b**) Experimental physical diagram.

**Figure 14 sensors-23-04895-f014:**
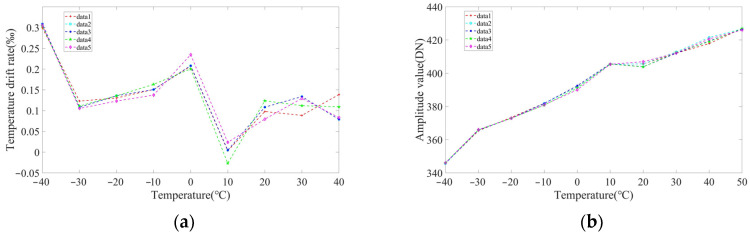
Detector temperature drift curve. (**a**) Temperature drift variation rate curve; (**b**) output variation curve with temperature.

**Table 1 sensors-23-04895-t001:** The influence of the number of accumulation periods on the static stability of the system.

Number of Accumulation Cycles	Mean Value	Variance	Standard Deviation
50	18,788	73.1	8.5
100	18,788	44.3	6.6
200	18,787	28.2	5.3

**Table 2 sensors-23-04895-t002:** The main parameters of the sensor.

Parameters	Value	Test Condition
Repeatability error	±0.62%	5 times
Nonlinearity	0.68%	5 times
Sensitivity	760 nm	-
Temperature stability	±0.00137% FS/°C	−40 °C to +50 °C
Temperature drift rate	<0.01%	Per 10 °C at room temperature

## Data Availability

Not applicable.

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
