# Peer review of "Wide Temperature Range and Low Temperature Drift Eddy Current Displacement Sensor Using Digital Correlation Demodulation"

_sensors, 2023, doi:10.3390/s23104895_

Round 1

Reviewer 1 Report

The experiments on sensor performance in this paper are comprehensive, but whether the sensor is superior to other sensors of the same type and its innovation need to be further explained.

1. The introduction should elaborate on the research status in this field. The references cited by the author are obviously insufficient, especially for the research on temperature drift sensors.

2. There is a problem with the description in lines 86-88. From formula (1), the distance between the sensor and the detection target directly affects the value of L2, and it is obvious that the relationship between L2 and the output voltage is nonlinear.

3. Please explain the derivation of formula (2).

4. Figure 5 is not very clear; in Figure 8 part of the content is blocked; The words in the annotation section in Figure 9 are too small.

5. Didn't the linearity test , the accuracy test and Temperature Stabilitybe repeated about 5 times? Why is there no error bar?

6. Please fully explain the innovation?

Reviewer 2 Report

The authors proposed a possible structure of eddy current sensors for low temperature drift. I don’t regard this manuscript contribute enough knowledge required by this journal. Also, it was written with many unclear parts, or something unproper. It is strange that a sensor with 5um range just has a resolution of 760nm, some thing wrong? What is the source of the orthogonal reference signals for digital correlation processing? The experimental result is lack of rigorous supporting materials.

Reviewer 3 Report

The paper presents an axial differential eddy current displacement sensor system to compensate for drift in the output signal due to environmental factors. These issues should be addressed by the authors to improve the manuscript.

1. Page 1, line 35: Please refine the sentence to be "An eddy current is induced within a nearby conductor by the alternating....". 

2. Page 3, Equation (2): Eq. 2 should be equal to zero if coils 1 and 2 are symmetrical to each other in terms of distance to the target and geometrical and electrical properties, and the induced voltages of U_R and U_L are in the same amplitude and phase. The authors should explain in the equation that the differential configuration is effectively canceling the signal contributed by temperature increase (180 degrees out of phase) and adding signals induced by the target (same phase).

3. Page 5 Eq. 14: what is "M"? the explanation is missing.

4. Page 8 Figure 6: How the data in Fig. 6 are obtained? This data should be from theoretical analysis of equations.

5. Page 8, line 220: Unit DN usage is unfamiliar to some readers. Please add a unit explanation of DN.

Reviewer 4 Report

The presented article is well described. The Authors described the theoretical basis and conducted experimental research. To fully understand the content, I am asking the Authors to explain a few issues and fill in the descriptive gaps:

- what is the assumption  "noise system is Gaussian white noise with a mean value of 0" (line 128)

- figure 7 is probably taken from the literature, a reference should be made to it

- the test stand should be described in more detail, e.g. by adding descriptions of the systems that were used for the measurements. Measurement parameters are also important. What sensors were used? Please add a measurement and data processing algorithm, also characteristic of the sensor.

- the novelty of the method should be emphasized in the article

- could the Authors add a figure of the test stand to the tasks completed in chapter 4.5?

- literature can be expanded: e.g. : a) about eddy current displacement sensor should analyze:  Wang and Ju (https://doi.org/10.1016/j.sna.2014.03.008 ) ; Zheng and Liu (https://doi.org/10.1109/JSEN.2019.2933347) ; Chaturvedi and Nabavi (https://doi.org/10.1109/JSEN.2017.2677526) b) please read about correlation analyses in articles by Lindstedt and Kulesza (https://doi.org/10.1155/2018/9134607);  Kulesza and Bartoszewicz (https://doi.org/10.1007/s11071-019-05221-0)  

Round 2

Reviewer 1 Report

The paper can be accepted, but the following revisions are still needed:

1. The case of Figure 1 should correspond to the formula.

2. There is a problem with the format of formula 1 and formula 4, please modify.

3. The two sets of folds in Figure 14 are not sufficiently differentiated. Please use two graphs for representation.

4. Please pay attention to consistent line formatting. The text in the image needs to be corrected with attention to case.

Reviewer 2 Report

Please clearify if the reference signals for demodulation synchronized with the exciting signal upon the coils.

Reviewer 4 Report

The Authors answered the questions asked by reviewer. The uncertainties were explained and described. In order to improve the quality of the article, Authors should also clarify the following issues in the article:

- add summary tables to test results. Despite the descriptions in the article, the reading before the recipient will be clearer

- include error analysis (also maybe in table)

- used  power density and correlation function formulas should also be supported by literature sources.

- is it possible to present in a clear form eg. in a block form the algorithm of the method (described in more detail than in Fig. 3)

Round 3

Reviewer 2 Report

No comments

Reviewer 4 Report

The Authors clarified the reviewer's doubts and answered all his questions. Relevant explanations have been updated in the article. The article can be accepted in its present form.